# Dynamic TMT-Based Quantitative Proteomics Analysis of Critical Initiation Process of Totipotency during Cotton Somatic Embryogenesis Transdifferentiation

**DOI:** 10.3390/ijms20071691

**Published:** 2019-04-04

**Authors:** Haixia Guo, Huihui Guo, Li Zhang, Yijie Fan, Yupeng Fan, Zhengmin Tang, Fanchang Zeng

**Affiliations:** State Key Laboratory of Crop Biology, College of Agronomy, Shandong Agricultural University, Tai’an 271018, China; diya_haixiaguo@163.com (H.G.); hhguo@sdau.edu.cn (H.G.); 15610418001@163.com (L.Z.); yjfan@sdau.edu.cn (Y.F.); fanyupeng@chnu.edu.cn (Y.F.); 17861500710@163.com (Z.T.)

**Keywords:** cotton, somatic embryogenesis, transdifferentiation, quantitative proteomics, regulation and metabolism, molecular basis, concerted network

## Abstract

The somatic embryogenesis (SE) process of plants, as one of the typical responses to abiotic stresses with hormone, occurs through the dynamic expression of different proteins that constitute a complex regulatory network in biological activities and promotes plant totipotency. Plant SE includes two critical stages: primary embryogenic calli redifferentiation and somatic embryos development initiation, which leads to totipotency. The isobaric labels tandem mass tags (TMT) large-scale and quantitative proteomics technique was used to identify the dynamic protein expression changes in nonembryogenic calli (NEC), primary embryogenic calli (PEC) and globular embryos (GEs) of cotton. A total of 9369 proteins (6730 quantified) were identified; 805, 295 and 1242 differentially accumulated proteins (DAPs) were identified in PEC versus NEC, GEs versus PEC and GEs versus NEC, respectively. Eight hundred and five differentially abundant proteins were identified, 309 of which were upregulated and 496 down regulated in PEC compared with NEC. Of the 295 DAPs identified between GEs and PEC, 174 and 121 proteins were up- and down regulated, respectively. Of 1242 differentially abundant proteins, 584 and 658 proteins were up- and down regulated, respectively, in GEs versus NEC. We have also complemented the authenticity and accuracy of the proteomic analysis. Systematic analysis indicated that peroxidase, photosynthesis, environment stresses response processes, nitrogen metabolism, phytohormone response/signal transduction, transcription/posttranscription and modification were involved in somatic embryogenesis. The results generated in this study demonstrate a proteomic molecular basis and provide a valuable foundation for further investigation of the roles of DAPs in the process of SE transdifferentiation during cotton totipotency.

## 1. Introduction

Somatic embryogenesis (SE) is a notable illustration of cell totipotency as one of the typical responses to abiotic stresses with hormone, which processes the developmental reprogramming of somatic cells toward the embryogenesis pathway. Cotton (*Gossypium hirsutum* L.), as the foremost natural fiber source [1] and one of the most important economic crops worldwide, has a global socioeconomic impact worth approximately $56 billion [2]. However, plant regeneration in SE is still a limiting method for transgenic development in cotton [1,3]. Somatic embryogenesis represents a unique phenomenon in the plant kingdom [4]. This developmental pathway is one of the most striking examples of plant cell developmental plasticity [5,6]. It includes a series of characteristic events, including somatic dedifferentiation, cell division activation, metabolism alterations and gene expression pattern reprogramming [4]. During SE, the development of somatic cells is reprogrammed to the embryogenic pathway, and SE forms the basis of cellular totipotency in higher plants [7]. Each transformed cell has the potential to produce a plant from the callus [8]. Somatic embryogenesis and subsequent plant regeneration have been reported in most major crop varieties [9]. Soybeans and cotton have proven to be the most difficult to regenerate [10]. 

In cotton, only a few percent of somatic embryos are able to mature and regenerate into plantlets. Most embryos develop abnormally, redifferentiate in to calli, or become necrotic and die [11]. Sakhanokho and Rajasekaran [12] obtained a variety of factors affecting cotton during in vitro regeneration, including plant growth regulators, explants, compositions of media and environmental conditions. The transition of somatic cells into embryogenic cells is the most intriguing and the least understood part of somatic embryogenesis [13,14,15]. Now, it is generally accepted that stress and hormones play a crucial role in collectively inducing cell dedifferentiation and initiation of the embryogenic program in plants with responsive genotypes [16,17,18]. 

Since the first observations of somatic embryo formation in suspension cultures of carrot cells by Stewards [7] and Reinert [19], the potential for SE has been demonstrated to be characteristic of extensive tissue culture systems from both dicotyledonous and monocotyledonous plants [20,21]. Considerable efforts have been expended in identifying the various factors that control SE [22,23]. An important gene that marks embryonic cells is the transcription factor gene *WUSCHEL* (*WUS*) [24]. Using a genetic gain-of-function screening approach, Zuo et al. [25] found that overexpression of *WUS* in roots, leaf petioles, stems, or leaves of *Arabidopsis* can induce the formation of somatic embryos. These results indicate that *WUS* participates in the promotion and/or maintenance of totipotent embryogenic stem cells. However, the *wus* mutants are still able to produce somatic embryos, suggesting that multiple alternative pathways can lead to the expression of totipotent potential. *WUS* is the only transcription factor that has been found to be involved in regulating meristematic stem cells (pluripotent) and embryogenic stem cells (totipotent) [15]. In addition, when auxin biosynthesis rates were manipulated in *Arabidopsis* embryos, polar auxin transport activity apparently buffered the normal distribution of auxin, suggesting a compensatory mechanism for buffering auxin gradients in the embryo, with *PIN1* and *PIN4* being the most important genes [26]. The results by Su [27] suggested that the establishment of auxin gradients and the polar distribution of *PIN1* are critical for the regulation of *WUS* expression during somatic embryogenesis. *ERF* plays an important role in hormone signal transduction and interconnecting different hormone pathways [28]. Inhibition of gibberellin (GA) biosynthesis increases the fraction of *lec1-tnp* seedlings displaying the mutant phenotype, suggesting that reduced GA levels enhance maturation processes induced by *LEAFY COTYLEDON 1* (*LEC1*) [29]. The plant hormone abscisic acid (ABA) regulates many important plant developmental processes and induces epigenetic reprogramming against tolerance to different stresses, including drought, salinity, low temperature and some pathogens [30,31]. ABA serves as a critical chemical messenger for stress responses. The roles of several genes in somatic embryogenesis masses (SEM) have been well-characterized, including *Arabinogalactan protein 1* (*AGP1*) [32], *Glutathione-S-transferase* (*GST*) [33], *SOMATIC EMBRYO RELATED FACTOR1* (*MtSERF1*) [34], *BABY BOOM* (*BBM*) [35], *Agamous-like 15* (*AGL15*) [36,37] and *SOMATIC EMBRYOGENESIS RECEPTOR-LIKE KINASE* (*SERK*) from *Daucus carota*, which was the first identified marker gene with a crucial role in SEM [38].

At present, a great number of SE-related genes and transcription factors have been identified at the transcription level. For example, Zeng [39], with the suppression subtractive hybridization (SSH) technique, identified 671 cDNAs in the initial period of SE in cotton. Nonetheless, reports on the identification of cotton SE at high-throughput proteins levels are still insufficient, especially during the initial stage of SE transdifferentiation. Proteomics is a powerful approach aimed at systematic studies of protein structure, function, interaction, and dynamics [4]. To further investigate the molecular regulatory mechanisms of somatic embryogenesis, the protein dynamics of NEC, PEC and GE were identified by TMT quantitative proteomics techniques. The highly sensitive proteomic platform based on the isobaric labels tandem mass tags was recently developed as one of the most robust proteomics techniques [40,41]. Through identification and annotation of DAPs, we uncovered the key genes/proteins and pathways involved in cotton SE transdifferentiation. The results generated in this study provide a valuable foundation for further investigation of the roles of DAPs in cotton SE.

## 2. Results

### 2.1. Somatic Embryogenesis in Cotton

PEC were formed from NEC after approximately 3 months of culture. The initial development period of GEs from embryogenic callus was the most restrictive step during cotton SEM (Figure 1). To identify proteins related to SE and morphogenesis in cotton, we sampled the critical representative periods of NEC, PEC and GEs for protein preparation and TMT-based quantitative proteomics analyses.

### 2.2. TMT-Based Quantitative Proteomic Basis Data Analysis and Overall Protein Identification

TMT-based quantitative proteomics was conducted to assess protein changes among NEC, PEC and GEs in cotton. Pair wise Pearson’s correlation coefficients displayed sufficient reproducibility of this experiment (Figure 2a). After quality validation, a total of 360,720 (74,579 matched) spectra were obtained. Of these spectra, 45062 identified peptides (27,673 unique peptides) and 9369 identified proteins (6730 quantified proteins) were detected (Table 1), and the average peptides mass error was <10 ppm, indicating a high mass accuracy of the MS data (Figure 2b). The lengths of most identified peptides were 8 to 20 amino acid residues (Figure 2c), suggesting that our sampling met the required standard. The detail information of identified proteins, including protein accession, protein description, gene name, peptide number, matching scores, carried charges and delta mass, is shown in Appendix A.

To further understand their functions, all identified proteins were annotated according to different categories, including subcellular localizations, Gene Ontology (GO) terms, Kyoto Encyclopedia of Genes and Genomes (KEGG) pathways, predicted functional domains and other data. The detailed information of all identified proteins is listed in Appendix A.

### 2.3. Enrichment of the Chloroplast Subcellular Location and GO Functional Classification of All Identified Proteins

To characterize the subcellular locations and functions of the identified differential proteins among NEC, PEC and GEs in cotton, subcellular locations and GO functional classification were performed (Figure 3; Appendix A). Subcellular distribution predictions (Figure 3a) showed that the identified proteins were distributed predominantly in chloroplast (31.10%), cytoplasm (25.75%) and nucleus (23.07%) during the transformation periods of somatic embryogenesis. Significantly, the highest proportion of differential proteins was enriched in chloroplasts, highlighting that this organelle plays an important role in cotton SE.

The results of cellular component analysis further revealed that 22.67% of the identified proteins were catalogued in organelles, 19.04% with macromolecular complexes and 18.07% with the membrane (Figure 3b). Regarding molecular function, the largest two GO categories, binding and catalytic activity, accounted for 47.59 and 41.55% of the identified proteins, respectively (Figure 3c). At the biological process level, proteins involved in the metabolic process, cellular process and single-organism process accounted for 33.17, 26.40 and 19.03% of identified proteins, respectively (Figure 3d). These results demonstrated that the identified proteins are found in multiple cellular components, have diversified molecular functions, and are involved in a variety of biological processes.

### 2.4. Identification of Differentially Abundant Proteins 

Differentially abundant proteins were defined as those with a ≥2-fold or ≤0.5-fold change in relative abundance (*p* < 0.05) between PEC and NEC, GEs and PEC, and GEs and NEC. In total, 805, 295 and 1242 DAPs were identified in comparing PEC versus NEC, GEs versus PEC and GEs versus NEC, respectively. In PEC compared with NEC, 805 proteins differentially accumulated were identified, 309 of which were up regulated and 496 of which were down-regulated. Of the 295 DAPs identified between GEs and PEC, 174 and 121 proteins were up- and down regulated in GEs, respectively. Of 1242 proteins differentially accumulated in GEs compared to NEC, 584 and 658 proteins were up- and down regulated, respectively (Figure 4a; Appendix A). 

To identify the commonly and specifically changed proteins between PEC and NEC, GEs and PEC or between GEs and NEC, a Venn diagram was generated (Figure 4b). It clearly showed that 122 and 29 proteins were specifically expressed in PEC and GE processes, respectively, and 85 common proteins (25 and 60) were involved in both PEC and GE.

To investigate the overall dynamics of proteome changes in SEM, we performed eight types of protein expression pattern analyses for the DAPs identified in NEC, PEC and GEs (Figure 4c). These analyses suggested protein expression patterns including down- to up regulation, up- to down regulation, down- to down regulation, up- to up regulation, down- to constant-regulation, up- to constant-regulation, constant- to down regulation and constant- to up regulation.

### 2.5. Enrichment Analysis of DAPs in GO, KEGG and Protein Domain

In total, 1418 proteins (nonrepetitive DAPs) were differentially accumulated and significantly regulated by the NEC, PEC and GEs under the given culture conditions (Appendix A/Total DAPs). The biological functions of the DAPs could also be identified by their GO terms, KEGG pathways and protein domain enrichment, as summarized in Figure 5, Figure 6 and Figure 7.

#### 2.5.1. Enrichment Cluster Analysis of DAPs between the Groups in GO Terms

In the different GO functional classifications, we carried out comparative cluster analysis between the sample groups, indicating the change of the co-expression trends of different proteins between the groups.

##### The Enzyme Metabolism Activity of Molecular Function Category in Cotton SE

For up-regulated DAPs, ‘protein dimerization activity’ and ‘protein heterodimerization activity’ showed a certain degree of enrichment in PEC versus NEC, GE versus PEC and GE versus NEC, especially in GE versus NEC. ‘DNA helicase activity’ and ‘helicase activity’ were functional categories which lower in GE versus NEC compared to PEC versus NEC. The up regulation of ‘peroxiredoxin activity’ and ‘nutrient reservoir activity’ in GE versus PEC was greater than in GE versus NEC, indicating that the enzyme and nutritional protein activities were higher in GE (Figure 5a). Most of the differential proteins are clearly clustered among the down regulated proteins. Four DAPs of peptidase-related protein activity, proteins from ‘hydrolase activity’ to ‘hydrolase activity, hydrolyzing O-glycosyl compounds’ and other enzyme activity were significantly down regulated in PEC versus NEC, but the enrichment of these DAPs was not significant in GE versus PEC; there were various degree of enrichment in GE versus NEC. Additionally, ‘glutamate dehydrogenase (NAD+) activity’, two proteins of ‘oxidoreductase activity’ and ‘phosphoenolpyruvate carboxykinase (ATP) activity’ accumulated to a certain extent in GE versus PEC (Figure 5a).

The results above indicated that enzyme metabolism activity affected the SE of cotton, with dynamic features in NEC, PEC, and GE.

##### The Photosynthesis-Related Proteins of the Cellular Component Category in Cotton SE

In the cellular component category of PEC versus NEC and GE versus PEC, a large number of DAPs were clustered in photosynthesis-related cellular components and proteins from ‘plastid thylakoid’ to ‘photosystem I’, indicating a significant decrease of photosynthesis in PEC versus NEC. Furthermore, in GE versus PEC, the corresponding photosynthetic cell components showed slightly up regulated enrichment. However, the photosynthetic effect of GE was not higher than NEC; this result showed that the photosynthesis-related DAPs of the cell component classification in GE versus NEC were concentrated in the down regulated expression region (Figure 5b). Additionally, the DAPs’ of expression pattern from ‘photosynthetic membrane’ to ‘photosystem II’, photosynthesis-related proteins’ was similar to the above results in that there was significant down regulation in PEC versus NEC (Figure 5b).

The analysis above showed that photosynthesis is a critical process involved in SE of cotton, which is consistent with our subcellular localization results.

##### The Regulation, Response and Metabolism-related Proteins of the Biological Process Category in Cotton SE

Proteins related to ‘lipid transport’, ‘reproductive system development’, ‘DNA metabolic process’ and ‘regulation’ were up regulated with different degrees of enrichment in PEC versus NEC and GE versus NEC. In addition, we also found that from ‘amide biosynthetic process’ to ‘cellular protein metabolic process’ proteins in GE versus NEC were uniformly enriched in up regulation (Figure 5c). Furthermore, ‘monosaccharide metabolic process’, ‘hexose metabolic process’, glycometabolism related proteins and the DAPs from ‘aminoglycan catabolic process’ to ‘cell wall macromolecule metabolic process’ were down regulated to different degrees in the three sample groups. Other proteins involved in response regulation were also enriched in the down regulated region in PEC versus NEC and GE versus NEC (Figure 5c).

Interestingly and consistently, the photosynthesis-related proteins in ‘photosynthesis, light harvesting’ were down regulated in PEC versus NEC and up regulated in GE versus PEC (Figure 5c). The above results demonstrated that SE of cotton might frequently involve proteins associated with environmental stress response, biological regulation, central metabolic processes, and photosynthetic metabolism.

#### 2.5.2. Enrichment Analysis in KEGG of the DAPs Involved in Phenylpropanoid Biosynthesis, Nitrogen Metabolism, Photosynthesis and Other Related Biological Processes

##### Enrichment Analysis in KEGG Clusters of Related Biological Processes among Groups

To further understand the function of SE-related proteins, we analyzed the differences and dynamic changes among groups of rich clustering classes in KEGG pathways. Cluster analysis of up regulated expression pathways showed that protein from ‘ribosome biogenesis in eukaryotes’ and ‘DNA replication’ were slightly enriched to varying degrees in PEC versus NEC and GE versus NEC (Figure 6a). In the down regulated enrichment region, multiple types of proteins were enriched in different levels in different sample groups, of which ‘phenylpropanoid biosynthesis’ and ‘nitrogen metabolism’ were enriched significantly in PEC versus NEC and GE versus PEC, respectively (Figure 6a). The categories ‘photosynthesis−antenna proteins’, ‘glycolysis/gluconeogenesis’ and ‘carbon fixation in photosynthetic organisms’ were down regulated in PEC versus NEC and up regulated in GE versus PEC (Figure 6a). 

Phenylpropanoid biosynthesis and nitrogen metabolism were significantly enriched and photosynthesis was re-enriched. The study suggesting the above biological processes possible involvement in cotton SE transformation. 

##### KEGG Pathway Enrichment Analysis of Related Biological Processes within the Sample Groups

In PEC versus NEC, KEGG pathway enrichment analysis demonstrated that the ‘phenylpropanoid biosynthesis’, ‘photosynthesis’, ‘glutathione metabolism’ and ‘glycolysis/gluconeogenesis’ were the most significantly affected pathways. A great deal of peroxidase proteins of the phenylpropanoid biosynthesis pathway were down regulated in PEC and NEC. Furthermore, a variety of proteins related to ‘photosynthesis’ and a slight number of DAPs of the ‘glycolysis/gluconeogenesis’ pathway were also down-regulated in PEC and NEC. Additionally, the ‘glutathione metabolism’ pathway was down regulated in PEC and NEC (Figure 6b). 

KEGG analysis of GE versus PEC indicated that the enriched pathways of the DAPs were most remarkably associated with ‘glutathione metabolism’ and ‘nitrogen metabolism’ pathways, which were both down regulated. Once again, ‘photosynthesis-antenna proteins’ and ‘photosynthesis’ were significantly enriched up regulated pathways (Figure 6c). 

In addition, DAPs of GE versus NEC were classified into various KEGG pathways, of which 6 metabolic pathways were significantly enriched. Interestingly, the largest number of DAPs was once more enriched in ‘phenylpropanoid biosynthesis’ and ‘photosynthesis’ were down-regulated. Moreover, ‘ribosome’ was up regulated pathways that were also enriched (Figure 6d). 

The results of the KEGG pathway analysis further indicated that the three metabolic pathways, including phenylpropanoid biosynthesis, nitrogen metabolism and photosynthesis, are essential for the process of cotton SE transformation.

#### 2.5.3. Enrichment Cluster Analysis of Differential Proteins Functional Domain 

Domain enrichment analysis of up regulated proteins revealed that ‘histone’-related domain, ‘ribosomal’-related domain, ‘seed maturation protein’, ‘translation protein SH3−like domain’, ‘RmlC−like jelly roll fold’, ‘Cupin 1’ and ‘RmlC−like cupin domain’ were enriched in the three sample groups with different dynamic expression patterns. The degree of enrichment of ’rmlC-like jelly roll fold’ and ‘cupin 1’, seed storage protein-related domain, was extremely high in GE versus PEC (Figure 7). This result indicates that the development of GE requires storage proteins to provide nutrients for the regeneration of somatic embryos. For the down regulated expression region, domains including the ‘aspartic peptidase domain’ and the ‘START−like domain’ were abundant in three or two samples groups, including glutathione S−transferase domain. Furthermore, domains related to ‘haem peroxidase, plant/fungal/bacterial’, ‘secretory peroxidase’, ‘aquaporin−like’ and ‘glycoside hydrolase’ were equally enriched in PEC versus NEC and GE versus NEC (Figure 7).

The cluster analysis of dynamic enrichment changes through different functional domains showed that the ‘rmlC−like cupin domain’, seed maturation protein, glutathione S−transferase and peroxidase-related domain were driving diverse tasks in different development processes of cotton SE.

### 2.6. Enrichment Analysis of the Major Biological Process between Different Comparison Groups

Above all, the significantly enriched GO terms of the biological process in different comparison groups were investigated. In PEC and NEC, the top 5 GO terms were peroxidase-related, further demonstrated the above results that significant enrichment of the phenylpropanoid biosynthesis pathway was observed during the initiation process of SE. In addition, ‘photosynthesis’ term showed significant changes in the PEC differentiation, promoting cotton SE. In GE and PEC, environmental response and photosynthesis related proteins were presented in the top 12 GO terms, indicating that the abundant abiotic stress and photosynthesis responsive proteins might regulate the maturity and development of globular embryos. In GE and NEC, the top 8 GO terms were peroxidase, photosynthesis and environmental response related proteins, being part of an important biological process in the process of cotton somatic embryo transformation. What’s more, glutamate dehydrogenase (GDH) is significantly enriched in the molecular function classification of PEC and GE. GDH is one of the main enzymes of nitrogen metabolism and participates in important biological processes of plant SE. These important biological processes throughout the development process suggested that complex regulatory networks are involved in the cotton SE process (Figure 8). 

### 2.7. Several Major DAPs are Associated with SE Regulation and Modification

SE of cotton is regulated by many factors. The GO and KEGG enrichment analysis showed that peroxidase, photosynthesis-related proteins, stress-responsive proteins, amino acid metabolism-related proteins and other energy metabolism enriched proteins all play an important role in the process of cotton SE. Furthermore, we comprehensively explored and analyzed differentially abundant proteins in cotton SE involving hormone signal response/signaling transduction, transcription/posttranscription and modification regulation (Table 2).

### 2.8. Comparative and Complementary Proteome of the Candidate DAPs

To complement the changes in abundance at the transcriptional level and confirm the authenticity and accuracy of the proteomic analysis, we analyzed ten candidate DAPs in NEC and PEC. Eight out of ten genes under this analysis showed positive correlation between the expression levels of protein and mRNA, indicating that most proteins were regulated directly at the transcription level. For the other two DAPs, negative correlation between their expression levels of protein and mRNA was observed, suggesting that their protein levels might be depended not only on the transcript level but also on the post-translational level (Figure 9).

## 3. Discussion

Proteomics analyses have long been recognized as a useful tool to dissect the molecular mechanisms of SE. The effectiveness of this technique is strongly dependent on the applied technique of the proteomics analysis system and the experimental system. Proteomics analyses for the somatic embryogenesis in *Pinus nigra Arn*. [42] and *Phoenix dactylifera* L. [43] have been previously performed. In this study, we performed quantitative proteomics analysis using the advanced EASY-nLC 1000 UPLC system based on the high-throughput TMT-labeling quantitative detection technique, and we consequently identified 9369 proteins for our samples. Thus, our new study significantly improved the resolution. TMT is advantageous and offers greater sensitivity for the analysis of cotton SE proteomic dynamics than previous methods. Furthermore, most previous studies have focused on the molecular mechanisms of regulation in the late stage of somatic embryogenesis, and little is known about protein regulation and metabolism in the early stage of embryogenesis. In the isobaric tags for relative and absolute quantitation (iTRAQ) proteomics analysis of Ge [11], 6318 proteins were identified in somatic spherical embryo and cotyledon embryo. Subsequently, Zhu [44] identified 5892 proteins related to SE through iTRAQ technology. In our study, a total of 9369 proteins were identified in stages of NEC, PEC and somatic embryos’ initial development period of GEs by TMT-labeling quantitative detection technique. Through identification and annotation of DAPs, we uncovered the key genes/proteins and pathways involved in the critical initial stage of cotton SE. The results generated in this study provide a valuable foundation for further investigation of the roles of DAPs during the expression of totipotency in cotton SE.

### 3.1. DAPs Enriched in Crucial Biological Processes Associated with Cotton SE

#### 3.1.1. Peroxidase Proteins Involved in Phenylpropanoid Biosynthesis Affect SE

Recently, a proteomic analysis of the somatic embryogenesis induction stage of *Medicago truncatula* revealed that peroxidase accumulates by day 5 after the induction of somatic embryogenesis and increases four-fold by day 14 [45]. Peroxidase is known to take part in diverse plant processes, such as auxin metabolism, cell wall elongation and stiffening [46]. However, in the present study, our data showed that peroxidase is involved in ‘phenylpropanoid biosynthesis’ pathways (Figure 6a,b; Figure 7), and the expression pattern was different from that of Almeida [45]. We presume that peroxidase might participate in different metabolic pathways with different expression patterns to regulate somatic embryogenesis in different species. Peroxidase is a phenol oxidase and is highly representative in date palm, and polyphenoloxidase are involved in oxidative browning in date palm [47,48]. Abohatem [49] demonstrates that the low rate of successive transfer culture (every 15 or 20 days) reduced the increase in phenolic contents and peroxidase activities in plant tissue leading to an enhancement of tissues/cells browning and then to a decrease in embryonic cell proliferation. Fresh culture medium every 7 days can significantly reduce the oxidative browning of tissue/cells, which is related to the reduction of phenolic compounds and peroxidase activity, thus increasing the proliferation of embryonic protocells. Based on our proteomic profile results, down regulation of many peroxidase DAPs in PEC versus NEC indicates that peroxidase protein activity is weak during SE, which prevents callus browning and promotes embryogenic differentiation.

#### 3.1.2. Photosynthesis in Cotton SE

The photosynthetic potential of cotyledon embryos has been reported in previous studies. For example, in *Coffea* × *arabusta* cotyledons photosynthetic capacity and germinated embryos, the cotyledon embryo stage is the earliest photoautotrophic stage to ensure plant development [50]. Rival [51] reported that the maximum photochemical activity of photosystem II is extremely low in proliferating embryos of oil palm and strongly increases at later developmental stages. In this study, the result is that the expression pattern of DAPs in the ‘photosynthetic’ pathway is down- to up regulation. According to our results, ‘Photosystem I P700 chlorophyll an apoprotein’, ‘oxygen-evolving enhancer protein 2’, ‘ATP synthase gamma chain’ and ‘Chlorophyll a–b binding protein’ of the ‘Photosynthesis’ pathway were down regulated in PEC in comparison to GEs (Figure 5b,c; Figure 6a–c). The results indicate that photosynthesis organs and photosynthetic capacity gradually developed from the GE stage for future autotrophy in cotton.

#### 3.1.3. Response to Environment Stresses during SE of Cotton

Stresses are the factors that have been increasingly recognized as having important role in the induction of SE [52]. Embryogenic competence of in vitro cultured somatic cells can be stimulated by various factors, such as phytohormone [53,54], dehydration [55,56], explant wounding [57], heavy metal ions [17,56], high osmotic pressure [58,59], etc. Our data showed that a series of stress responsive biological processes were significantly enriched in PEC and GE, including response to acid chemical, water, inorganic substance, oxygen-containing compound, chemical, abiotic stimulus and biotic stimulus, indicating that callus was protected from the external environmental stress through complex regulatory networks to ensure the embryos development of cotton during the PEC-to-embryos transition of cotton (Figure 5c; Figure 8b). Previous studies have shown that decrease of water availability stimulated a shift from proliferation of cells and early embryos to the production of cotyledonary embryos in the developmental program of the culture [60]. In our study, the water content gradually decreased with the callus culture time increased prompted the tissue cells to respond to dry stress and initiate defense mechanisms to ensure embryo development. In the carrot somatic embryogenesis, 2,4-dichlorophenoxyacetic acid (2,4-D) functions as a stress chemical as well as an auxin [61]. During Arabidopsis somatic embryos induction, cells in the shoot apical meristem (SAM) of wild-type seedlings acquired pluripotency or embryogenic potential under initial stress, and then these cells form somatic embryos on 2,4-D treatment [17]. In our culture system, 2,4-D is also used to induce SE, but stress treatment is necessary before exposure to 2,4-D. From the datas above, we can conclude that stress responses are the indispensable biological processes in plant embryos induction.

#### 3.1.4. Effects of Nitrogen Metabolism Related to SE

Glutamate dehydrogenase (GDH) is one of the enzymes directly related to nitrogen metabolism. Induction of various GDH isoenzymes may suggest their varied anabolic and catabolic functions [62]. GDH, which is directly involved in the oxidation of amino acids, protect tissues from the toxicity of ammonium [63]. Ganced [64,65] strongly suggested that sugar could control the expression of the GDH gene through catabolic inhibition, which has been described in bacteria and yeast. In this study, GDH in nitrogen metabolism was significantly enriched in GE versus PEC and showed a down regulated trend (Figure 6a,c). We speculate that GDH may be more important during the cotton PEC redifferentiation period.

### 3.2. Other DAPs of Regulatory Factors Associated with Cotton SE

#### 3.2.1. Phytohormone Response Related Proteins

Hormones are the most likely candidates in the regulation of developmental switches [16]. Auxin is the main growth regulator in plants, which is involved in the regulation of cell division and differentiation, as well as the existence of other growth regulators such as abscisic acid [66], ethylene [67], gibberellin [68]. 

In cotton primary embryogenic calli, the DAPs of ‘Auxin efflux carrier component’ (AP2) and ‘indole-3-acetic acid-amido synthetase GH3. 17-like’ (GH3) were up regulated (Table 2). Auxin-related proteins are essential for initiating dedifferentiation and cell division in already differentiated cells before they can express embryogenic competence [44]. The *PIN* gene is believed to be the coding element that regulates the auxin efflux mechanism of the polar auxin transport, which is concluded by the polarity localization of the PIN membrane protein and auxin absorption experiment [69]. Blilou [70] pointed to polar auxin transport as a major factor in organ formation. The *GH3* gene is one of several sequences screened by differential hybridization of auxin-induced cDNA sequences extracted from auxin-treated soybean tissues [71]. Expression of the *GH3* gene has been shown to be rapidly and specifically induced by the application of auxin [72,73]. These conclusions also explained the up regulation of AP2 and GH3 in PEC due to the trend of auxin polar transport organ formation and the key regulation of induced somatic embryo formation.

Gallie [74] identified two ethylene receptor gene families in maize. In developing embryos, the expression levels of members of the two ethylene receptor families were significantly increased, which indicated that embryonic development was involved in ethylene synthesis. In this study, DAPs of ‘ethylene receptor-like isoform X1’ (Table 2) were involved in the ethylene signaling pathway and were up regulated in PEC compared to NEC, suggesting that ethylene receptor may positively regulate SE initiation in cotton.

In the present study, the differentially accumulated ‘GA-stimulated transcript-like protein 1’ (GASL1) was down regulated in PEC and GE (Table 2). Ge [11] demonstrated that, at 10 days after GA treatment, 95% of the embryos showed an aberrant structure, large size, and light-green color. Therefore, we presume that GA negatively affects somatic embryo production and growth via regulation of the GA signaling pathway.

Treatment with ABA improves the efficiency of somatic embryo maturation of *Panax ginseng* [75] and promotes sugar cane embryo growth [76]. Somatic embryos treated with ABA generate the highest yield of plantlets in *Picea abies* [77]. In our results, the ‘abscisic acid receptor PYR1-like’ DAP, which is involved in the ABA signal pathway, was up regulated in GE compared to PEC (Table 2); this suggests that the ‘abscisic acid receptor PYR1-like protein’ promotes embryo maturation in GE.

#### 3.2.2. Signal Transduction Related Proteins

In this study, we also found that a large number of important signal regulators, including kinases, calcium signals and GTP-binding related proteins (Table 2), were significantly differential expressed in cotton SE. Ca^2+^ has an important role in the establishment of cellular polarity during embryogenesis in plants [78]. It as a secondary messenger may trigger various signal transduction pathways [79] in plants SE. Calmodulin or Ca^2+^-dependent protein kinase may be involved in the regulation of Ca^2+^ levels in the proembryogenic cell mass (PEM) of sandalwood to promote embryo development [80]. In our TMT profile, the calcium-related proteins ‘calcium-dependent protein kinase 11-like’ and ‘probable calcium-binding protein CML27’ were simultaneously up regulated in PEC versus NEC. We presume that calcium-related proteins involved in signal transduction and regulation of calcium balance, thereby establishing polarity to promote cotton SE.

Phosphoenolpyruvate carboxylase (PEPC), a glycolytic protein and CO_2_-fixing enzyme [81], participates in TCA cycle under non-photosynthesis conditions [82] and signal transduction of plant embryo development [83]. Therefore, this metabolic pathway maintains the carbon residue pool necessary for oil and storage protein biosynthesis that occurred the later in embryonic development [84]. In our study, the protein ‘phosphoenolpyruvate carboxykinase [ATP]-like’ was up regulated in PEC versus NEC and down regulated in GE versus PEC (Table 2). Thus, we presume that PEPC may not only participate in TCA cycle and fixation of CO_2_ but also in signal transduction in PEC, providing energy for cotton SE transformation process.

As a result, calcium-related proteins, PEPC and other signal transduction proteins may be involved in a variety of biological processes to promote the transformation process of cotton SE.

#### 3.2.3. SE Associated Proteins of Aquaporins and Fatty Acid Metabolism

In our TMT profile, a large number of aquaporins, including PIP and TIPs, were found to be significantly involved in PEC and GE initiation during SE. Aquaporin was involved in water transport by osmosis to prevent dryness and abortion during embryonic development, which was of great significance in the development of *Picea asperata* somatic embryos [85]. Here, we found that the aquaporins (PIP210, TIP14 and PIP type) were significantly down regulated during PEC; the aquaporins (PIP2-5, TIP3-2) were significantly up regulated during GE (Table 2). Therefore, we suspect that these proteins are sensitive to water content and light induction during cotton SE initiation process of transdifferentiation. Similarly, previous studies have shown that aquaporin, under stress conditions, forms a ‘tunnel’ to regulate the water transport in the cell membrane [86], and can be induced by light [87]. In addition to reducing the activation energy of water transport, aquaporin also enhanced the permeability of the plasma membrane [88].

Many fatty acid biosynthesis- and metabolism-related proteins, such as ‘cytochrome P450 86B1-like’ and ‘cytochrome P450 86A8-like’, were differentially accumulated. In the fatty acid biosynthesis pathway, they were up regulated in PEC (Table 2). A previous study reported that fatty acids, which affect cell function and growth patterns, appear to be a part of the TDZ action pattern and may play an important role in inducing regeneration [89]. These results implied that aquaporins and perturbations of fatty acid metabolism contribute to the initiation of SE in cotton.

#### 3.2.4. Regulation of SE-related Proteins, Transcription, Posttranscription and Modification

In this study, two types of SE-related proteins and multiple types of transcription factors, zinc finger domains, microRNAs and modification-related proteins were identified (Table 2). Late embryogenesis abundant protein (LEA) was first found in cotton (*Gossypium hirsutum*) seeds and accumulated in the late stages of embryogenesis, which played a crucial role in cell dehydration tolerance [90]. In current study, two types of SE-related proteins, ‘embryonic protein DC-8-like’ and ‘embryogenesis abundant protein’, were up regulated in GE. It indicating that they may positively regulate somatic embryo maturation and involve in cell dehydration tolerance in cotton. 

*WUSCHEL* (*WUS*) is a vital transcription factor for labeling embryonic cells [24]. The capability of promoting the vegetative to embryonic transition by *WUS*, and eventually forming somatic embryos, suggesting that the homeodomain protein also plays a critical role during embryogenesis, in addition to its function in meristem development [91]. In this result, ‘*WUSCHEL*-related homeobox 9-like’ was up regulated in PEC, indicating that the *WUSCHEL*-related protein is essential for the initiation of somatic embryogenesis. In addition to *WUS*, we also found other transcription factors, zinc finger domains related to SE (Table 2).

Members of the Argonaute protein family are key players in the small RNA-directed gene silencing pathway. Various types of small RNA and Argonaute proteins played important roles in embryonic development, cell differentiation and transposon silencing in all higher eukaryotes [92]. In our data, ‘protein argonaute 1-like isoform X2’, ‘protein argonaute 4-like’ and ‘small RNA 2’-O-methyltransferase-like isoform X4’ were significantly up regulated during PEC versus NEC (Table 2), suggesting that RNA-mediated post-transcriptional regulation played an important role in the process of cotton SE transformation as reported in other plants [93,94,95].

In current study, we identified several types of modification-related DAPs, including DNA methylation, chromatin modification, acetylation, ubiquitination and phosphorylation. In the process of carrot SE, the removal of auxin led to the loss of DNA methylation, so that the embryo continued to develop [96]. Six types of methylation-related proteins were identified, which dynamically regulated to cotton SE by different expression patterns. Efficient modification of chromatin structure was crucial in the epigenetic regulation of genes [97]. In our data, ‘chromatin modification-related protein MEAF6-like isoform X3’ promoted maturation and development of cotton GEs by epigenetic regulation. In addition, during the PEC period, we also identified acetylation, ubiquitination and phosphorylation-related proteins with different expression patterns (Table 2), indicating that they played diverse and indispensable functions in regulating cotton SE.

The different expression patterns of SE-related proteins, multitudinal transcription, posttranscription and modification showed that they played a pivotal role in the process of cotton SE [98,99,100,101,102,103]. The results of our high-through put proteomics assay, the large scale of proteins associated with SE, and their complex expression patterns suggest that SE is a concerted process involving multiple molecular pathways controlled by a complicated gene regulatory network.

## 4. Materials and Methods 

### 4.1. Plant Materials and Culture Conditions

Seeds of cotton cultivar Yuzao 1 (Institute of Cotton Research of CAAS), uncovering the coats, were imbibed in 0.1% (*w*/*v*) HgCl_2_ for 10 min, then rinsed four times by sterile distilled water and germinated on Murashige and Skoog (MS) medium containing 3% (*w*/*v*) sucrose and 0.3% (*w*/*v*) Phytegel. Hypocotyl explants (0.5–1.0 cm) taken from 5–7 d old seedlings and non-embryogenic callus initiation of Yuzao 1 was done, as described by Wu [104], in MS medium plus B5 vitamins medium (MSB) containing 0.46 µmol L–1 kinetin and 0.45 µmol L–1 2,4-D. Calli were subcultured in MSB medium without any hormone to induce embryogenic calli (EC), as described by Zeng [39]. Samples were collected from the following three stages of SEM (1) NEC from explants cultured in MSB, induced (2) PEC and (3) GEs. The collected samples were immediately frozen in liquid nitrogen, and stored at −80 °C before protein extraction. Each staged sample was prepared for three biological replicates.

### 4.2. Protein Extraction and Identification

#### 4.2.1. Protein Samples Preparation

The three biological replicates of sample were first ground by liquid nitrogen, then the powder was transferred to 5 cm^3^ centrifuge tube and sonicated three times on ice using a high intensity ultrasonic processor (Scientz, Ningbo, China) in 4-fold volume phenol extraction buffer (including 10 mM dithiothreitol, 1% Protease Inhibitor Cocktail and 2 mM EDTA). The equal amount of trisaturated phenol (pH 8.0) was added; then, the mixture was further vortexed for 5 min. After centrifugation (4 °C, 10 min, 5000× *g*), the upper stage of phenol was transferred to a new centrifuge tube. Proteins were precipitated by adding at five volume of 0.1 M ammonium sulfate-saturated methanol to precipitate overnight. After centrifugation at 4 °C for 10 min, the supernatant was discarded. The remaining precipitate was washed with ice-cold methanol once, followed by ice-cold acetone for three times. The protein was redissolved in 8 M urea and the protein concentration was determined with BCA kit (Beyotime Biotechnology, Shanghai, China) according to the manufacturer’s instructions.

#### 4.2.2. Trypsin Digestion

Protein solution with dithiothreitol makes its final concentration of 5 mM, 56 °C for 30 min. After that, acetamide was added to make the final concentration 11 mM and incubated in the dark at room temperature for 15 min. Finally, the sample urea concentration was diluted to less than 2 M. With 1:50 (*w*/*w*) quality ratio (trypsin: protein) to join the pancreatic enzyme, 37 °C enzyme solution for the night. Then the trypsin (Promega, Madison, WI, USA) was added at a mass ratio of 1:100 (trypsin: protein), and the enzymatic hydrolysis continued for 4 h.

#### 4.2.3. TMT Labeling

After trypsin digestion, the peptide was desalted by Strata X C18 SPE column (Phenomenex, Torrance, CA, USA) and vacuum-dried. The peptide was reconstituted in 0.5 M TEAB (Sigma, St. Louis, MI, USA) and processed according to the manufacturer’s protocol for TMT kit (Thermo Fisher Scientific, Waltham, MA, USA). Briefly, one unit of TMT reagent was thawed and reconstituted in acetonitrile. The peptide mixtures were then incubated for 2 h at room temperature and pooled, desalted and dried by vacuum centrifugation. 

#### 4.2.4. HPLC Fractionation and LC-MS/MS Analysis

The tryptic peptides were fractionated into fractions by high pH reverse-phase HPLC using Agilent 300Extend C18 column (5 μm particles, 4.6 mm ID, 250 mm length, Agilent, Santa Clara, USA). Briefly, peptides were first separated with a gradient of 8% to 32% acetonitrile (pH 9.0) over 60 min into 60 fractions. Then, the peptides were combined into 9 fractions and dried by vacuum centrifuging.

The tryptic peptides were dissolved in 0.1% formic acid (solvent A), directly loaded onto a home-made reversed-phase analytical column (15-cm length, 75 μm i.d.). The gradient was comprised of an increase from 8% to 23% solvent B (0.1% formic acid in 90% acetonitrile) over 38 min, 23% to 35% in 14 min and climbing to 80% in 4 min then holding at 80% for the last 4 min, all at a constant flow rate of 450 nL/min on an EASY-nLC 1000 UPLC system (Thermo Fisher Scientific, Waltham, MA, USA).

The peptides were subjected to NSI source followed by tandem mass spectrometry (MS/MS) in Q Exactive^TM^ HF-X (Thermo Fisher Scientific, Waltham, MA, USA) coupled online to the UPLC. The electrospray voltage applied was 2.0 kV. The m/z scan range was 350–1600 *m*/*z* for a full scan, and intact peptides were detected in the Orbitrap (Thermo Fisher Scientific, Waltham, MA, USA) at a resolution of 60,000. Peptides were then selected for MS/MS using NCE setting as 28 and the fragments were detected in the Orbitrap at a resolution of 17,500. A data-dependent procedure that alternated between one MS scan followed by 20 MS/MS scans with 30 s dynamic exclusion. Automatic gain control (AGC) was set at 1E5. The fixed first mass was set as 100 *m*/*z*. The HPLC Fractionation and LC-MS/MS Analysis in our research is supported by Jingjie PTM BioLabs (Hangzhou, China).

#### 4.2.5. Database Search

The resulting MS/MS data were processed using Maxquant search engine (v.1.5.2.8). Tandem mass spectra were searched against the UniProt 14.1 (2009)—*Gossypium hirsutum* database (78,387 sequences) concatenated with reverse decoy database. Trypsin/P was specified as cleavage enzyme allowing up to 2 missing cleavages. The first search was 20 ppm, the main search was 5 ppm, and the fragment ion mass tolerance was 0.02 Da. 

### 4.3. Bioinformatics 

#### 4.3.1. Annotation Methods

##### GO Annotation

Gene Ontology (GO) annotation proteome was derived from the UniProt-GOA database (http://www.ebi.ac.uk/GOA/). Firstly, converting identified protein ID to UniProt ID and then mapping to GO IDs by protein ID. If some identified proteins were not annotated by UniProt-GOA database, the InterProScan soft would be used to annotated protein’s GO functional based on protein sequence alignment method. Then proteins were annotated according to the biological process, cellular component and molecular function of the three categories of protein Gene Ontology annotation.

##### Domain Annotation 

Identified proteins domain functional description were annotated by InterProScan (a sequence analysis application) based on protein sequence alignment method, and the InterPro domain database was used. InterPro (http://www.ebi.ac.uk/interpro/) is a database that integrates diverse information about protein families, domains and functional sites, and makes it freely available to the public via Web-based interfaces and services. At the heart of the database are diagnostic models, known as signatures, from which protein sequences can be searched for potential functions. InterPro has applications in large-scale genome-wide and metagenomic analyses, and the characterization of individual protein sequences.

##### KEGG Pathway Annotation

KEGG connects known information on molecular interaction networks, such as pathways and complexes (the “Pathway” database), information about genes and proteins generated by genome projects (including the gene database) and information about biochemical compounds and reactions (including compound and reaction databases). These databases are different networks, known as the “protein network”, respectively, and the “chemical universe”, respectively. Efforts are being made to increase KEGG knowledge, including information on orthographic clustering in the KEGG orthographic database. KEGG Pathways mainly including: Metabolism, Genetic Information Processing, Environmental Information Processing, Cellular Processes, Rat Diseases, Drug Development. Kyoto Encyclopedia of Genes and Genomes (KEGG) database (http://www.genome.jp/kegg/) was used to annotate protein pathway. Firstly, using KEGG online service tools KAAS (http://www.genome.jp/kaas-bin/kaas_main) to annotate protein’s KEGG database description. Then mapping the annotation result on the KEGG pathway database using KEGG online service tools KEGG mapper (http://www.kegg.jp/kegg/mapper.html).

##### Subcellular Localization Prediction

Eukaryotic cells are elaborately subdivided into membrane-bound chambers with unique functions. Some major constituents of eukaryotic cells are extracellular space, cytoplasm, nucleus, mitochondria, Golgi apparatus, endoplasmic reticulum (ER), peroxisome, vacuoles, cytoskeleton, nucleoplasm, nucleolus, nuclear matrix, and ribosomes. There, we used wolfpsort (http://www.genscript.com/psort/wolf_psort.html) a subcellular localization predication soft to predict subcellular localization. Wolfpsort is an updated version of PSORT/PSORT II for the prediction of eukaryotic sequences. 

#### 4.3.2. Functional Enrichment

##### Enrichment of Gene Ontology Analysis

Through GO annotation, proteins are divided into biological process, cellular compartment, and molecular function. For each class, we used a double-tailed Fisher’s precision test to detect the enrichment of differentially abundant proteins relative to all identified proteins. GO with a revised *p* value of <0.05 is considered significant.

##### Enrichment of Pathway Analysis

KEGG database identified enrichment pathways by double-tailed Fisher’s precision test to detect the enrichment of differentially abundant proteins against all identified proteins. The pathway with a corrected *p*-value < 0.05 was considered significant. According to the KEGG website, these paths are classified into hierarchical categories.

##### Enrichment of Protein Domain Analysis

For each category protein, InterPro (a resource that provides a functional analysis of protein sequences by classifying them into families and predicting the presence of domains and important sites) database was researched and a two-tailed Fisher’s exact test was employed to test the enrichment of the differentially abundant proteins against all identified proteins. Protein domains with a *p*-value < 0.05 were considered significant.

##### Enrichment-Based Clustering

For further hierarchical clustering based on different protein functional classification (such as GO, Domain, Pathway, Complex). We first collated all the categories obtained after enrichment along with their *p* values, and then filtered for those categories which were at least enriched in one of the clusters with *p* value <0.05. This filtered P value matrix was transformed by the function x = −log10 (*p* value). Finally, these x values were z-transformed for each functional category. These z scores were then clustered by one-way hierarchical clustering (Euclidean distance, average linkage clustering) in Genesis. Cluster membership was visualized by a heat map using the “heatmap.2” function from the “gplots” R-package.

## Figures and Tables

**Figure 1 ijms-20-01691-f001:**
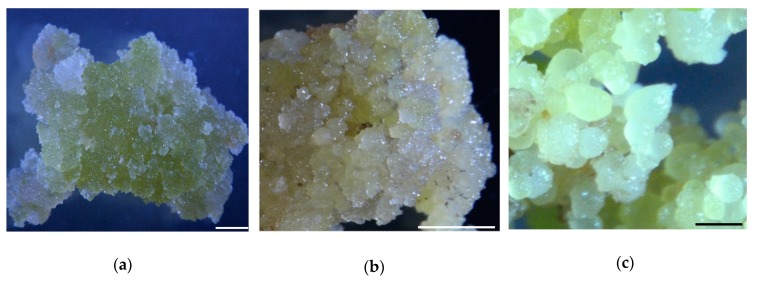
Samples used for proteomic assays: (**a**) Nonembryogenic calli; (**b**) Primary embryogenic calli; (**c**) Globular embryos. Bar (**a**,**b**) = 2.5 mm; bar (**c**) = 0.5 mm.

**Figure 2 ijms-20-01691-f002:**
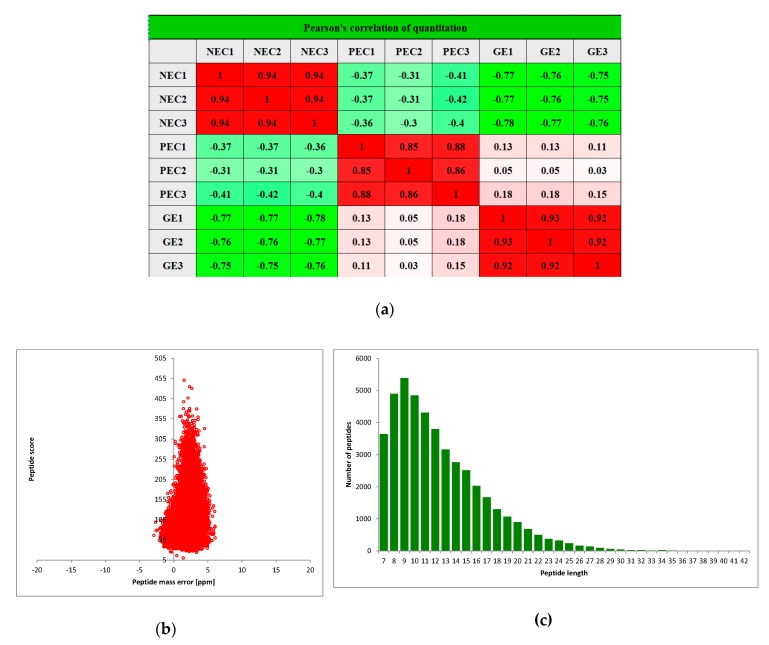
Experimental strategy for quantitative proteome analysis and quality control validation of MS data: (**a**) Mass delta of all identified peptides; (**b**) Average peptide mass error; (**c**) Length distribution of all identified peptides. NEC: Nonembryogenic calli; PEC: Primary embryogenic calli; GE: Globular embryos. Each staged sample was prepared for three biological replicates.

**Figure 3 ijms-20-01691-f003:**
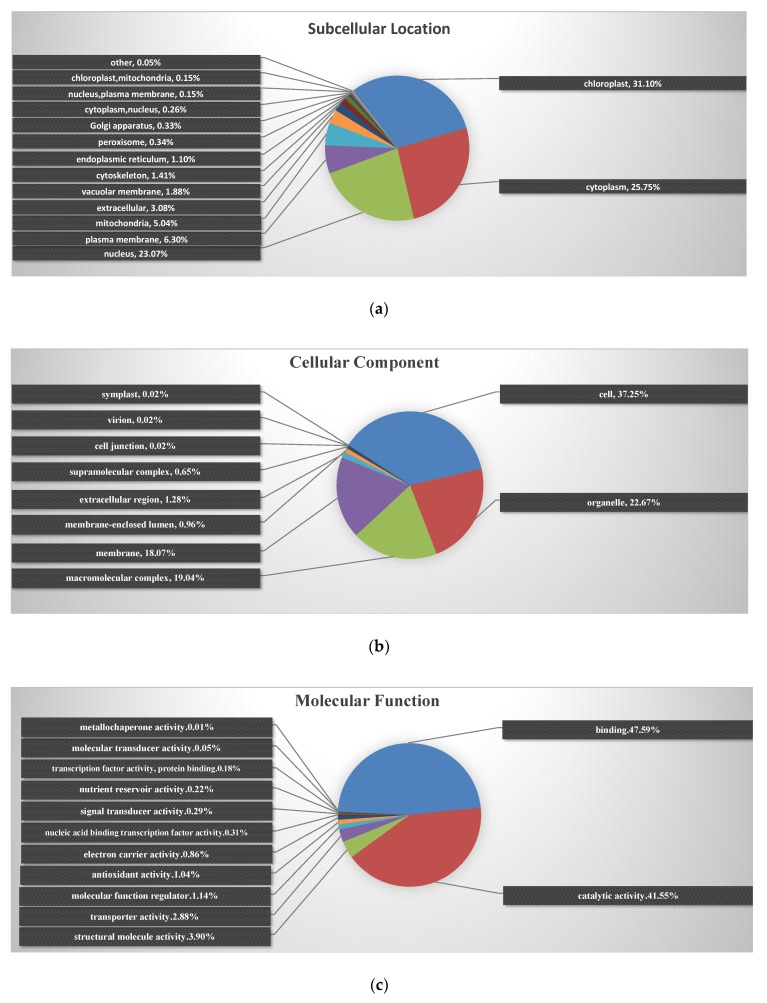
Subcellular functional annotation and GO functional classification of identified proteins. (**a**) Subcellular locations of identified proteins; (**b**) GO annotation in terms of cellular component; (**c**) GO annotation in terms of molecular function; (**d**) GO annotation in terms of biological process. GO: Gene Ontology.

**Figure 4 ijms-20-01691-f004:**
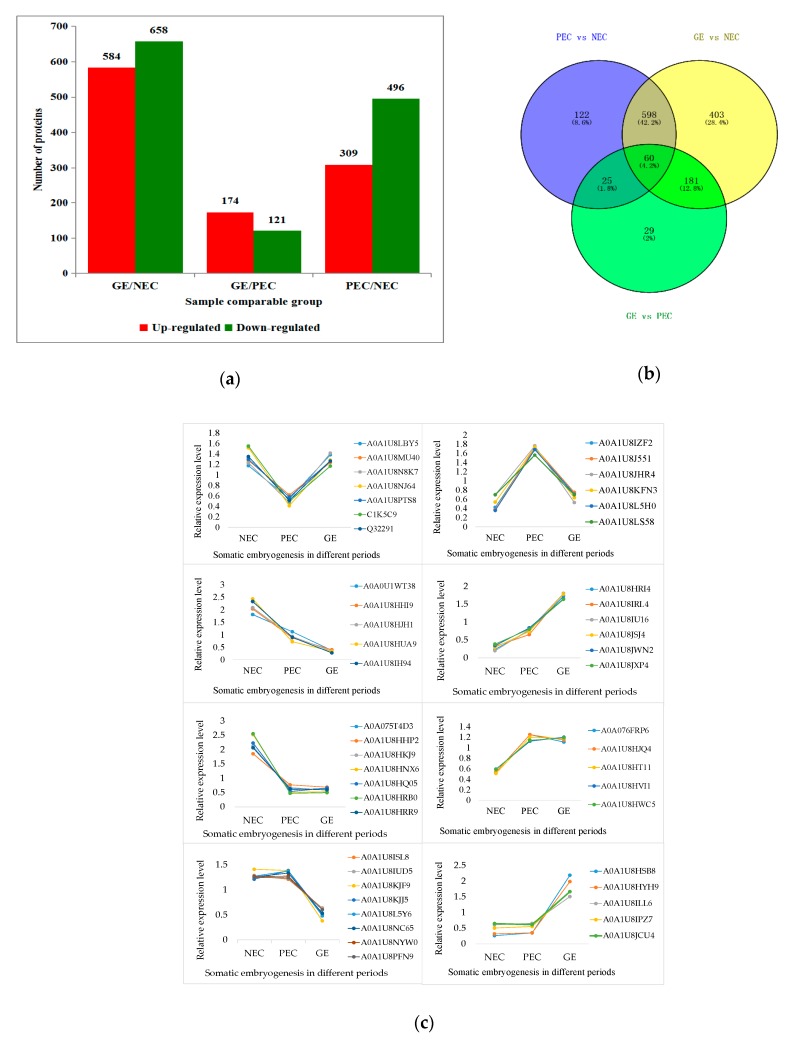
Distribution of differentially accumulated proteins (DAPs): (**a**) Number of up(red)- and down(green)-regulated DAPs in GE vs. NEC, GE vs. PEC and PEC vs. NEC; (**b**) Venn diagram to show the distribution of DAPs between PEC vs. NEC (blue circle), GE vs. PEC (yellow circle) and GE vs. NEC (green circle). (**c**) Expression patterns of DAPs. NEC: Nonembryogenic calli; PEC: Primary embryogenic calli; GE: Globular embryos.

**Figure 5 ijms-20-01691-f005:**
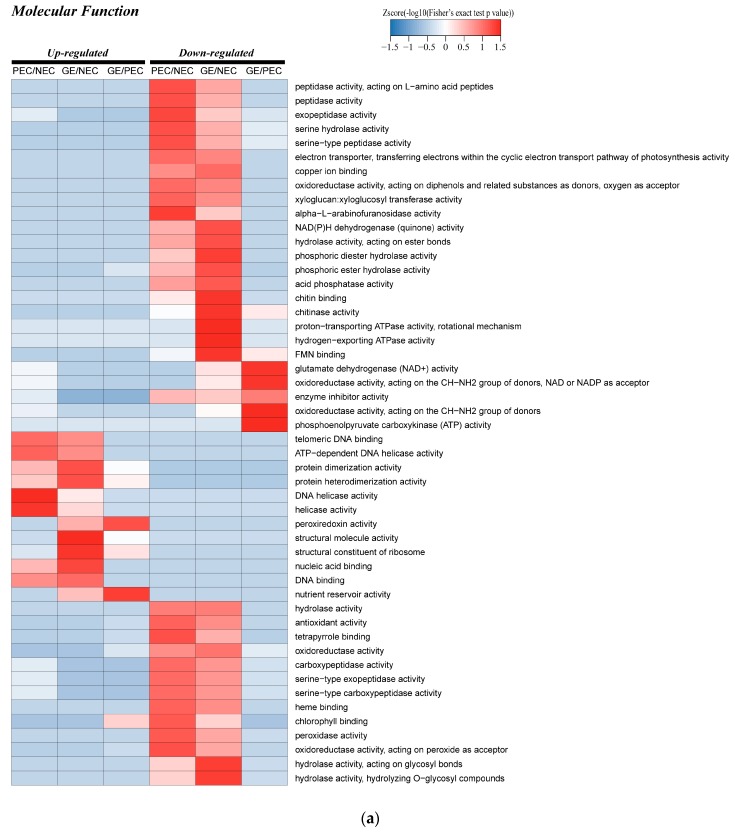
GO functional cluster of differentially accumulated proteins (DAPs) in PEC vs. NEC, GE vs. PEC and GE vs. NEC: (**a**) GO functional cluster of DAPs in the molecular function; (**b**) GO cluster of DAPs in the cellular component; (**c**) GO functional cluster of DAPs in the biological process. NEC: Nonembryogenic calli; PEC: Primary embryogenic calli; GE: Globular embryos; GO: Gene Ontology.

**Figure 6 ijms-20-01691-f006:**
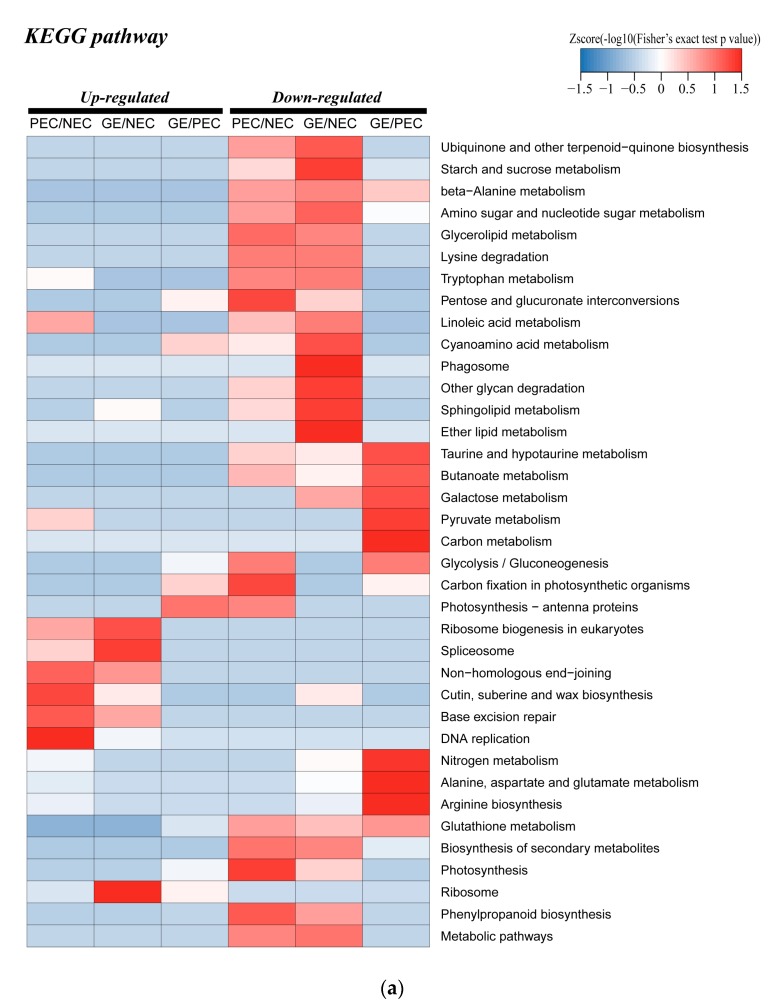
KEGG cluster and pathway enrichment analysis of DAPs: (**a**) KEGG clusters in PEC vs. NEC, GE vs. PEC and GE vs. NEC; (**b**) Pathway enrichment in PEC vs. NEC; (**c**) Pathway enrichment in GE vs. PEC; (**d**) Pathway enrichment in GE vs. NEC. The pathway enrichment statistical analysis was performed by Fisher’s exact test. The X-axis is folded enrichment; the y-axis is enrichment pathway. The mapping is the protein number. NEC: Nonembryogenic calli; PEC: Primary embryogenic calli; GE: Globular embryos; KEGG: Kyoto encyclopedia of genes and genomes

**Figure 7 ijms-20-01691-f007:**
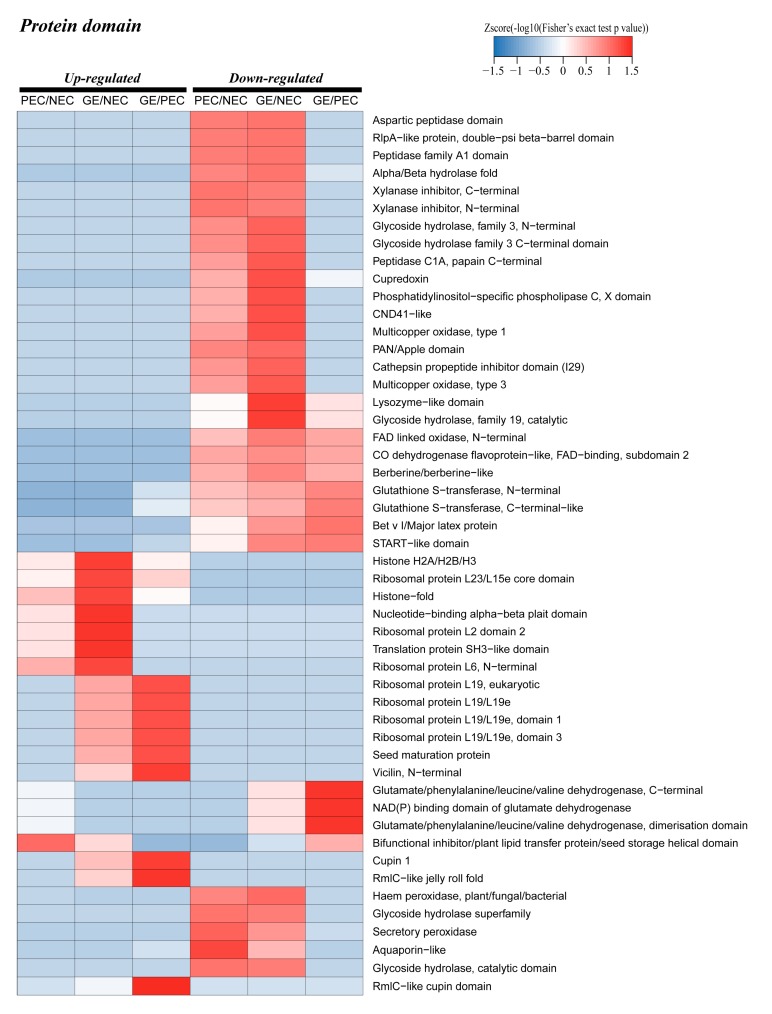
Protein domain enrichment analysis of the differentially accumulated proteins (DAPs) in PEC vs. NEC, GE vs. PEC and GE vs. NEC. NEC: Nonembryogenic calli; PEC: Primary embryogenic calli; GE: Globular embryos.

**Figure 8 ijms-20-01691-f008:**
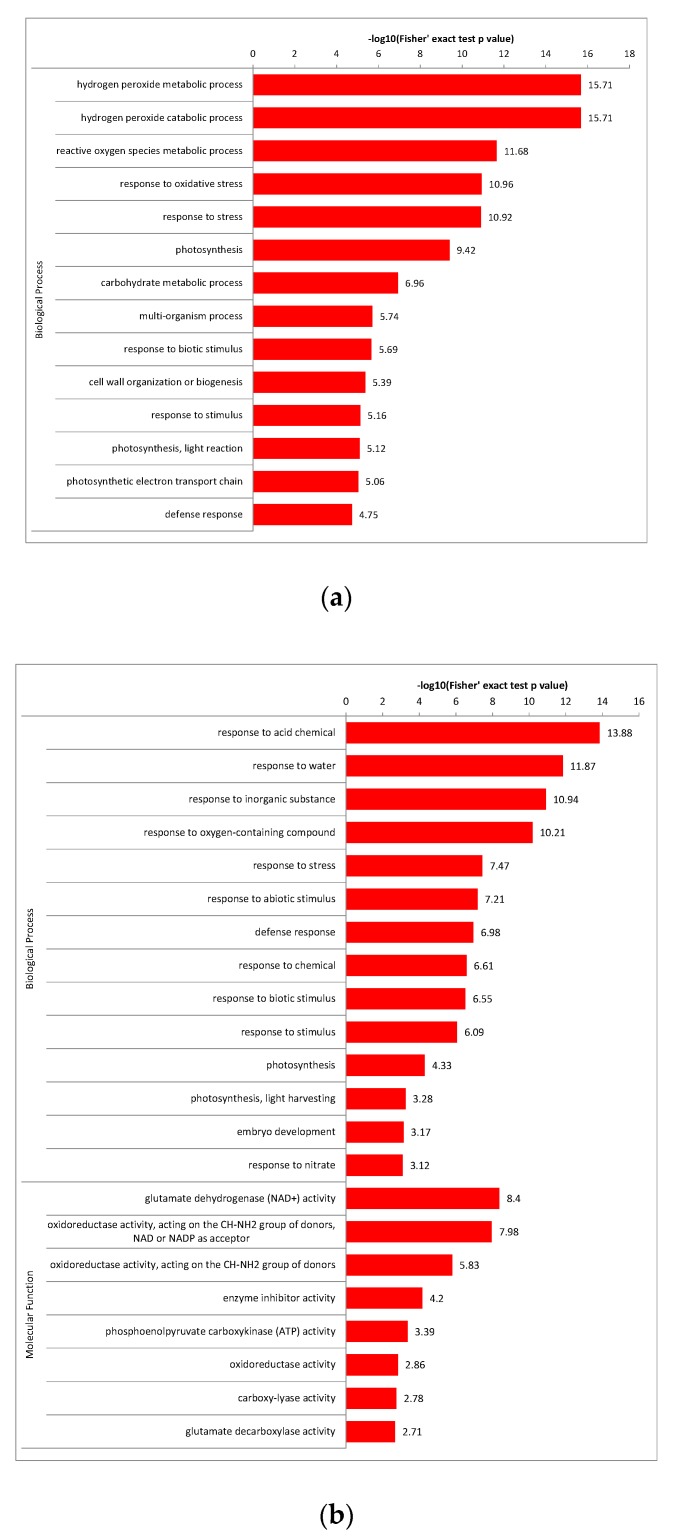
GO terms of differentially accumulated proteins (DAPs) in the biological process and the molecular function: (**a**) The biological process of DAPs in PEC vs. NEC; (**b**) The biological process and the molecular function of DAPs in GE vs. PEC; (**c**) The biological process of DAPs in GE vs. NEC. NEC: Nonembryogenic calli; PEC: Primary embryogenic calli; GE: Globular embryos; GO: Gene Ontology.

**Figure 9 ijms-20-01691-f009:**
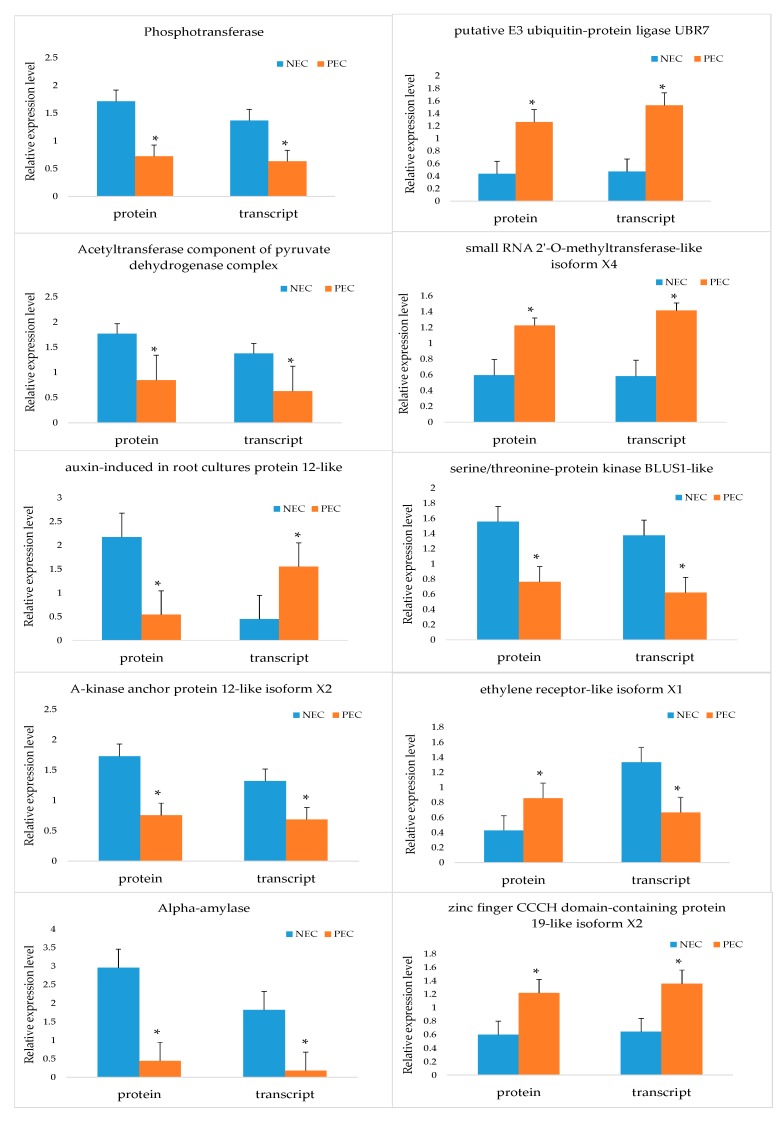
Comparative and complementary proteome of the candidate DAPs in stage of NEC and PEC. Significant differences in expression level were indicated by “*”. DAPs: differentially accumulated proteins; NEC: Nonembryogenic calli; PEC: Primary embryogenic calli; GE: Globular embryos.

**Table 1 ijms-20-01691-t001:** MS/MS spectrum database search analysis summary.

Total Spectrum	Matched Spectrum	Peptides	Unique Peptides	Identified Proteins	Quantifiable Proteins
360,720	74,579 (20.7%)	45,062	27,673	9369	6730

**Table 2 ijms-20-01691-t002:** Significantly representative SE regulatory DAPs in PEC vs. NEC, GE vs. PEC and GE vs. NEC.

Gene ID	Gene Name	Protein ID	Protein Description	Pathway Annotation	PEC/NEC Ratio	GE/PEC Ratio	GE/NEC Ratio
LOC107907377	PIN2	A0A120KAE0	Auxin efflux carrier component	Auxin signal	2.36	—	—
LOC107909506	GH3.17	A0A1U8JQJ4	Indole-3-acetic acid-amido synthetase GH3.17-like isoform X2	Auxin signal	2.14	—	—
LOC107948437	ETR1	A0A1U8NHA4	ethylene receptor-like isoform X1	Ethylene signal	2.02	—	3.705
LOC107938108	GASL1	M1GN42	GA-stimulated transcript-like protein 1	GA signal	0.25	0.27	0.068
LOC107955576	GASL4	M1GMV2	GA-stimulated transcript-like protein 4	GA signal	3.00		—
LOC107950128	PYR1	A0A1U8NR07	Abscisic acid receptor PYR1-like	ABA signal	—	2.31	2.361
LOC107893363	At5g01020	A0A1U8I6G4	serine/threonine-protein kinase At5g01020-like	Signal transduction	0.50	—	0.482
LOC107897915	—	A0A1U8IRF6	A-kinase anchor protein 12-like isoform X2	Signal transduction	0.44	—	0.459
LOC107909143	—	A0A1U8JP78	leucine-rich repeat receptor-like protein kinase PXC2	Signal transduction	0.45	—	0.36
LOC107945188	BAM3	A0A1U8N9I0	leucine-rich repeat receptor-like Ser/Thr -protein kinase BAM3	Signal transduction	2.03	—	—
LOC107935259	PCKA	A0A1U8M980	phosphoenolpyruvate carboxykinase [ATP]-like	Signal transduction	2.08	0.38	—
LOC107943515	At1g56140	A0A1U8N4H5	probable LRR receptor-like serine/threonine-protein kinase At1g56140	Signal transduction	0.29	—	0.215
LOC107931208	TPK1	A0A1U8LYM7	thiamine pyrophosphokinase 1-like isoform X1	Signal transduction	0.46	—	0.469
LOC107905700	PFK	A0A1U8JGW8	ATP-dependent 6-phosphofructokinase	Signal transduction	—	2.55	2.504
LOC107943957	PV42A	A0A1U8N623	SNF1-related protein kinase regulatory subunit gamma-like PV42a	Signal transduction	—	2.83	2.757
LOC107937641	CPK11	A0A1U8MGW7	calcium-dependent protein kinase 11-like	Signal transduction	2.25	—	3.84
LOC107930954	CML27	A0A1U8LUL1	probable calcium-binding protein CML27	Signal transduction	4.16	—	2.88
LOC107916423	RHN1	A0A1U8KFK5	ras-related protein RHN1-like	Signal transduction	—	0.47	0.333
LOC107889787	—	A0A1U8HV05	Embryonic protein DC-8-like	Somatic embryogenesis related proteins	—	3.58	4.522
LOC107937048	Lea2A-A	Q03791	Embryogenesis abundant protein	Somatic embryogenesis related proteins	—	4.42	4.746
LOC107941722	WOX9	A0A1U8MVD7	WUSCHEL-related homeobox 9-like	Transcription factor	2.58	—	—
LOC107905698	NFYB6	A0A1U8JC47	Nuclear transcription factor Y subunit B-6	Transcription factor	3.22	2.07	6.661
	bHLH4	W5XUY9	BHLH4 transcription factor	Transcription factor	2.16	—	—
LOC107920272	NFYB9	A0A1U8KSD1	nuclear transcription factor Y subunit B-9-like	Transcription factor	4.07	—	2.509
LOC107931333		A0A1U8LVZ2	transcription factor HBP-1b (C38)-like	Transcription factor	2.40	—	—
LOC107924015	PHL1	A0A1U8L8P3	Protein PHR1-LIKE 1-like	Transcription factor	0.15	—	0.166
LOC107891610	At1g07170	A0A1U8I119	PHD finger-like domain-containing protein 5B	Zinc finger	3.49	—	4.21
LOC107909066	NERD	A0A1U8JUI6	zinc finger CCCH domain-containing protein 19-like isoform X2	Zinc finger	2.03	—	—
LOC107927097	TAF15B	A0A1U8LG36	transcription TFIID subunit 15b-like	Zinc finger	0.44	—	—
LOC107962890	ZHD5	A0A1U8PUW9	zinc-finger homeodomain protein 5-like	Zinc finger	—	2.93	5.554
LOC107890886	AGO1	A0A1U8HY77	protein argonaute 1-like isoform X2	Posttranscriptional regulation	8.44	—	6.734
LOC107906203	AGO4	A0A1U8JEA7	protein argonaute 4-like	Posttranscriptional regulation	2.38	—	2.426
LOC107962954	HEN1	A0A1U8PXD5	small RNA 2’-O-methyltransferase-like isoform X4	Posttranscriptional regulation	2.06	—	—
LOC107891032	IDM1	A0A1U8HYR9	increased DNA methylation 1-like isoform X4	Modification-related protein	2.62	0.49	—
LOC107906306	MMT1	A0A1U8JEK1	Methionine S-methyltransferase	Modification-related protein	0.46	—	0.458
LOC107948568	SUVH4	A0A1U8NHS0	histone-lysine N-methyltransferase, H3 lysine-9 specific SUVH4-like isoform X2	Modification-related protein	0.28	—	0.196
LOC107943854	CCOAOMT	A0A1U8N5R4	caffeoyl-CoA O-methyltransferase -like	Modification-related protein	0.35	—	0.363
LOC107953938	EMB1691	A0A1U8P3T9	methyltransferase-like protein 1	Modification-related protein	2.33	—	2.774
LOC107916882	IAMT1	A0A1U8KGS5	indole-3-acetate O-methyltransferase 1	Modification-related protein	2.32	—	—
LOC107958653	—	A0A1U8PI31	chromatin modification-related protein MEAF6-like isoform X3	Modification-related protein	—	3.48	5.641
LOC107926365	—	A0A1U8LDK7	RNA cytidine acetyltransferase	Modification-related protein	2.17	—	2.333
LOC107960303	—	A0A1U8PLN7	Acetyltransferase component of pyruvate dehydrogenase complex	Modification-related protein	0.48	—	0.404
LOC107947121	UBR7	A0A1U8NDR8	putative E3 ubiquitin-protein ligase UBR7	Modification-related protein	2.90	—	2.679
LOC107959749	RUB2	A0A1U8PJK7	ubiquitin-NEDD8-like protein RUB2	Modification-related protein	0.26	—	0.286
LOC107938100	—	A0A1U8MII2	Phosphotransferase	Modification-related protein	0.42	—	0.476
LOC107898863	CYP86B1	A0A1U8IP72	Cytochrome P450 86B1-like	Fatty acid	4.51	—	2.388
LOC107922796	CYP86A8	A0A1U8L513	Cytochrome P450 86A8-like	Fatty acid	2.02	—	—
LOC107915850	PIP2-5	A0A1U8KIL6	probable aquaporin PIP2-5	Aquaporins	—	2.58	—
LOC107898442	TIP3-2	A0A1U8IU16	probable aquaporin TIP3-2	Aquaporins	—	2.29	9.086
LOC107963873	GhPIP2;10	D8FSK4	Aquaporin PIP210	Aquaporins	0.14	—	0.102
—	GhTIP1;4	D8FSK6	Aquaporin TIP14	Aquaporins	0.20	—	0.195
LOC107934987 LOC107944588	PIP1;4	G8XV51	PIP protein	Aquaporins	0.22	—	0.109

DAPs: differentially accumulated proteins; NEC: Nonembryogenic calli; PEC: Primary embryogenic calli; GE: Globular embryos; GO: Gene Ontology.

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
