# Peer review of "Dynamic TMT-Based Quantitative Proteomics Analysis of Critical Initiation Process of Totipotency during Cotton Somatic Embryogenesis Transdifferentiation"

_ijms, 2019, doi:10.3390/ijms20071691_

Round 1

Reviewer 1 Report

Dear Authors,

Reviewer comments ijms-459555

The mansucript entitled „Dynamic TMT-based quantitative Proteomics analysis of critical initation process of totipotency during cotton somatic embryogenesis transdifferentiation“ represents a useful study aimed at an investigation of proteome in nonembryogenic calli (NEC), primary embryogenic calli (PEC), and globular embryos (GEs) during somatic embryogenesis in cotton.

I have several minor, but important comments on the manuscript.

1/ Terminology:

Use the term „differentially abundant proteins“ instead of „differentially expressed proteins“  since proteomic analysis determines only alterations in protein relative abundance which reperesents always a result between protein biosynthesis („protein expression“) and protein degradation.

Use rather the term „data complementation“ than „data validation“ for transcriptome data compared to proteome data since both transcripts and proteins can reveal differential dynamics in their levels under the same conditions.

2/ Results:

In Figure 1, a scale bar providing information about calli size has to be added to the microphotographs.

A scheme summarising major differences at proteome level (major protein groups and their alterations) between nonembryogenic calli (NEC), primary embryogenic calli (PEC), and globular embryos (GEs) of cotton during somatic embryogenesis process should be added as a figure at the end of Results.

In Supplementary materials, appropriate data to each identified protein have to be provided including MS score, number of unique peptides including sequences of all matched unique peptides including appropriate statistics have to be given.

3/ Materials and methods:

The source of plant material used for the experiments, i.e., cotton cultivar Yuzao 1, has to be specified. The authors have to write from which institution (Research company, gene bank, etc.), the seeds were obtained.

Protein identification, Database search:

The database parameters of the UniProt-Gossypium hirsutum database concatenated with reverse decoy database have to be specified including the database version (date of release), number of protein sequences and residues.

Units: Use SI units for all volume items, i.e., „5 cm3“ instead of „5 mL“ (line 465), i.e., „cm3“ instead of „mL“, „mm3“ instead of „μL“, „μm3“ insetad of „nL.“

Materials and methods, line 4.3.1.4. Subcellular localization, I do not understand the second paragraph about bacteria since no bacteria were investigated in the study. I think that this paragraph should be omitted (deleted).

Further formal comments:

Abstract, line 23: „five differentially abundant proteins“ (not „five differentially expressed proteins“).

Abstract, line 28: Use rather the verb „complement“ than „validate“ in the sentence „We also used previous transcriptome data (unpublished) to complement the authenticity and accuracy of the proteomic analysis.“

Abstract, line 32: Add a space between the words „a“ and „proteomic molecular basis“, i.e., „a proteomic Molecular basis“.

Introduction, line 53: Use rather the word „percents“ than „frequencies“ in the sentence „In cotton, only a few percents of somatic embryos are able to mature and regenerate into plantlets.“

Introduction, line 54: Add a space between the words „differentiate“ and „in“, i.e., „differentiate in to calli“

Introduction, line 57: Add a space between the words „cells“ and „is“

Results, line 168: Add a space between the words „and 85 common proteins“.

Results, line 203: Add a space between the words „extent“ and „in.“

Results, line 284: Add a space between the words „high“ and „in.“

Discussion, line 444: Use the term “differentially accumulated“ insetad of „differentially expressed.“

Discussion, line 446: Add „a“ preceding the words „part of the TDZ action pattern“, i.e., „appear to be a part of the TDZ action pattern…“

Add a space between the words „can express“.

Line 535: Add a comma preceding the word „respectively“ in the sentence „…and the „chemical universeů, respectively.“

Final recommendation: Reconsider after a major revision.

Author Response

Mar. 26, 2019

IJMS

We are returning our manuscript (ijms-459555) entitled “Dynamic TMT-based quantitative proteomics analysis of critical initiation process of totipotency during cotton somatic embryogenesis transdifferentiation” that has been revised based on the reviewer's suggestions. We appreciate the time and expertise of the reviewers and the suggestions helped us to improve our manuscript.

Here are our point by point response to the reviewer's comments:

Reviewer 1

Comments to the Author:

The mansucript entitled “Dynamic TMT-based quantitative Proteomics analysis of critical initiation process of totipotency during cotton somatic embryogenesis transdifferentiation” represents a useful study aimed at an investigation of proteome in nonembryogenic calli (NEC), primary embryogenic calli (PEC), and globular embryos (GEs) during somatic embryogenesis in cotton.

I have several minor, but important comments on the manuscript.

Terminology

1. Use the term “differentially abundant proteins” instead of “differentially expressed proteins”, since proteomic analysis determines only alterations in protein relative abundance which represents always a result between protein biosynthesis (“protein expression”) and protein degradation.

RESPONSES: We thank the reviewer for the comments and suggestions. The term “differentially abundant proteins” has been used instead of “differentially expressed proteins” in the revised manuscript.

2. Use rather the term “data complementation” than “data validation” for transcriptome data compared to proteome data since both transcripts and proteins can reveal differential dynamics in their levels under the same conditions.

RESPONSES: We thank and agree with the reviewer’s comments. The term “data complementation” has been used instead of “data validation” in the revised manuscript.

Results

3. In Figure 1, a scale bar providing information about calli size has to be added to the microphotographs.

RESPONSES: We thank the reviewer for pointing out the mistake there. The scale bars have been added to the microphotographs.

4. A scheme summarizing major differences at proteome level (major protein groups and their alterations) between nonembryogenic calli (NEC), primary embryogenic calli (PEC), and globular embryos (GEs) of cotton during somatic embryogenesis process should be added as a figure at the end of Results.

RESPONSES: We thank the reviewer’s comments and suggestions. A scheme summarizing major differences at proteome level between the three stages during cotton SE process has been added as a figure at the end of Results section.

5. In Supplementary materials, appropriate data to each identified protein have to be provided including MS score, number of unique peptides including sequences of all matched unique peptides including appropriate statistics have to be given.

RESPONSES: We agree with the reviewer’s suggestions. The detailed information of each identified protein has been provided in Supplementary Table 1.

Materials and methods

6. The source of plant material used for the experiments, i.e., cotton cultivar Yuzao 1, has to be specified. The authors have to write from which institution (Research company, gene bank, etc.), the seeds were obtained.

RESPONSES: We thank the reviewer for the comments and suggestions. The institution source of Yuzao 1 seeds has been added in the revised manuscript.

7. Protein identification, Database search: The database parameters of the UniProt-Gossypium hirsutum database concatenated with reverse decoy database have to be specified including the database version (date of release), number of protein sequences and residues.

RESPONSES: We thank and agree with the reviewer’s comments. The database parameters, including version (date of release), number of protein sequences and residues, have been specified in the text.

8. Units: Use SI units for all volume items, i.e., “5 cm3” instead of “5 mL” (line 465), i.e., “cm3” instead of “mL”, “mm3“ instead of “μL”, “μm3” insetad of “nL”.

RESPONSES: We agree with the reviewer’s suggestions. All volume items have been used SI units in the revised manuscript.

9. Materials and methods, line 4.3.1.4. Subcellular localization, I do not understand the second paragraph about bacteria since no bacteria were investigated in the study. I think that this paragraph should be omitted (deleted).

RESPONSES: We thank and agree with the reviewer’s comments and suggestions. The paragraph about bacteria has been deleted.

Further formal comments

10. Abstract, line 23: “five differentially abundant proteins” (not “five differentially expressed proteins”).

Abstract, line 28: Use rather the verb “complement” than “validate” in the sentence “We also used previous transcriptome data (unpublished) to complement the authenticity and accuracy of the proteomic analysis”.

Abstract, line 32: Add a space between the words “a” and “proteomic molecular basis”, i.e., “a proteomic Molecular basis”.

Introduction, line 53: Use rather the word “percents” than “frequencies” in the sentence “In cotton, only a few percents of somatic embryos are able to mature and regenerate in to plantlets”.

Introduction, line 54: Add a space between the words “differentiate” and “in”, i.e., “differentiate in to calli”.

Introduction, line 57: Add a space between the words “cells” and “is”.

Results, line 168: Add a space between the words “and 85 common proteins“.

Results, line 203: Add a space between the words “extent” and “in”.

Results, line 284: Add a space between the words “high” and “in”.

Discussion, line 444: Use the term “differentially accumulated” insetad of “differentially expressed”.

Discussion, line 446: Add “a” preceding the words “part of the TDZ action pattern”, i.e., “appear to be a part of the TDZ action pattern…”

Add a space between the words “can express”.

Line 535: Add a comma preceding the word “respectively” in the sentence “…and the chemical universe, respectively”.

RESPONSES: We thank the reviewer for pointing out the formal mistakes there. The sentences have been revised to be correct accordingly. Simultaneously, we also checked and corrected other formal mistakes in the whole article.

Reviewer 2 Report

The work of Guo et al. provides an important amount of data about proteins present in three different initial stages of somatic embryogenesis in cotton. Although authors compare the abundance of these proteins between these three stages, the data is presented in a fuzzy manner with no clear focus about the biological process(es) that could be related to the initial steps of somatic embryogenesis. Authors expose the raw results obtained from several annotation databases with no clear nomenclature associated to the identified proteins. Furthermore, authors make conclusions about a key role of certain genes/proteins identified in the study just mentioning the name of them and the assignated categorization by databases used without looking on the published literature if these genes/proteins has been previously described or have a role in somatic embryogenesis. Finally, authors overlook the literature related to the topic in cotton (and other related species) at the moment of generate a fruitful discussion about the initial steps of somatic embryogenesis which is one of the most important and interesting points of high throughput analysis (transcriptomic, proteomic, metabolic, etc).

I strongly recommend to authors to consolidate the data obtained from the different annotation databases used to analyze the data, make a deeper analysis of the enriched categories looking for genes/proteins described in the literature, even if no related to somatic embryogenesis and construct a fruitful discussion related to the possible role of these genes/proteins in the initial step of somatic embryogenesis in cotton (offering a perspective of their signifcance for other plant species too). Also, I will recommend to the authors to avoid the use of unpublished data to make any kind of analysis if this data is not present in any repository where readers can access, otherwise this kind of approach attemps against scientific reproducibility. Finally, I suggest to the authors to revise carefully the material and methods section, complete the information related to the manufacturer of reagents and equipment used in this study and eliminate any extra information unrelated to the procedures performed to obtain the results presented in the work. After all these changes, I would expect that this dataset and manuscript can offer a significant contribution to the field.

Author Response

Mar. 26, 2019

IJMS

We are returning our manuscript (ijms-459555) entitled “Dynamic TMT-based quantitative proteomics analysis of critical initiation process of totipotency during cotton somatic embryogenesis transdifferentiation” that has been revised based on the reviewer's suggestions. We appreciate the time and expertise of the reviewers and the suggestions helped us to improve our manuscript.

Here are our point by point response to the reviewer's comments:

Reviewer 2

Comments to the Author:

1. The work of Guo et al. provides an important amount of data about proteins present in three different initial stages of somatic embryogenesis in cotton. Although authors compare the abundance of these proteins between these three stages, the data is presented in a fuzzy manner with no clear focus about the biological process(es) that could be related to the initial steps of somatic embryogenesis. Authors expose the raw results obtained from several annotation databases with no clear nomenclature associated to the identified proteins. Furthermore, authors make conclusions about a key role of certain genes/proteins identified in the study just mentioning the name of them and the assignated categorization by databases used without looking on the published literature if these genes/proteins has been previously described or have a role in somatic embryogenesis. Finally, authors overlook the literature related to the topic in cotton (and other related species) at the moment of generate a fruitful discussion about the initial steps of somatic embryogenesis which is one of the most important and interesting points of high throughput analysis (transcriptomic, proteomic, metabolic, etc).

RESPONSES: We thank the reviewer for the valuable comments and suggestions. As the reviewer recommend, we have revised the manuscript accordingly. Firstly, the biological processes related to the initial steps of somatic embryogenesis have been focused on phenylpropanoid biosynthesis, photosynthesis, response to environment stresses and nitrogen metabolism in the modified sections (results and discussion). Besides, clear nomenclatures associated to the differentially abundant proteins have been added in Table 2. And we have also amended the format and content in the two Supplementary tables. Furthermore, we thoroughly rewritten and improved the discussion by looking on a series of published literatures on somatic embryogenesis. In details, we have fully discussed each type of important differentially abundant proteins, and analyzed and compared its important role in cotton somatic embryogenesis with the previous published related literatures. Finally, we greatly agree with and implement the reviewer's comment that a series of the important related literatures need to be cited and comparatively analyzed in the discussion about the initial steps of somatic embryogenesis.

 2. I strongly recommend to authors to consolidate the data obtained from the different annotation databases used to analyze the data, make a deeper analysis of the enriched categories looking for genes/proteins described in the literature, even if no related to somatic embryogenesis and construct a fruitful discussion related to the possible role of these genes/proteins in the initial step of somatic embryogenesis in cotton (offering a perspective of their signifcance for other plant species too). Also, I will recommend to the authors to avoid the use of unpublished data to make any kind of analysis if this data is not present in any repository where readers can access, otherwise this kind of approach attemps against scientific reproducibility. Finally, I suggest to the authors to revise carefully the material and methods section, complete the information related to the manufacturer of reagents and equipment used in this study and eliminate any extra information unrelated to the procedures performed to obtain the results presented in the work. After all these changes, I would expect that this dataset and manuscript can offer a significant contribution to the field.

RESPONSES: We acknowledge and agree with the reviewer’s comments. The entire manuscript has been modified as the reviewer recommend. Firstly, the discussion section have been revised and reorganized to deeply analyze the possible role of the enriched proteins with a series of the published relevant literatures during the initial step of somatic embryogenesis in cotton and other plant species. Also, we highly agree with the reviewer's opinion and the transcriptome data will be uploaded to the GeneBank database where readers can access soon. Finally, we have revised the Material and Methods section carefully as the reviewer suggested. The corresponding information associated with the manufacturer of reagents and equipment used in this study have been completed. We also removed and added the related procedures presented in the modified section.

We hope you find the revised manuscript satisfactory and thank you for considering our manuscript.

Sincerely,

Fanchang Zeng

Professor

State Key Laboratory of Crop Biology

College of Agronomy,

Shandong Agricultural University

Tai’an 271018

Phone: +86-538-8241828

E-mail: fczeng@sdau.edu.cn

Round 2

Reviewer 1 Report

Dear Authors,

Reviewer comments ijms-459555.R1 revised

The revised manuscript entitled „Dynamic TMT-based quantitative Proteomics analysisof critical initiation process of totipotency during cotton somatic embryogenesis transdifferentiation“ was significantly improved by the authors with respect to the original version.

However, I still have one major comment and several formal comments on the revised manuscript.

1/ Major comment: In Table 2 providing basic data on identified proteins revealing differential abundance between PEC vs NEC and GE vs PEC, respectively, proteins revealing differential abundance between GE vs NEC have to be added. I do not understand why proteins revealing differential abundance between GE vs NEC were not included in Table 2. I think that they have to be added there.

2/ Formal comments:

I still have several formal comments on the revised manuscript.

Results:

Line 312: Correct the typing error in the word „investigated“ (not „invesgated“).

Line 317: Modify the term „photosynthesis responsive proteins“ (NOT „photosynthesis responded proteins“).

Line 324: Add the verb „are“ preceding the verb „involved“ in the sentenc e „…suggested that complex regulatory networks are involved in the cotton SE process…“

Discussion:

Line 506: Correct the typing error in the word „involve“ (not „invole“).

Line 523: Correct the verb form „led“ (not „leaded“) in the sentence „In the process of carrot SE, the removal of auxin led to the loss of DNA methylation so that the embryo continued to develop…“

Materials and methods,

Line 570: Modify the verb form in the verb „were ground“ (not „was grinded“) in the sentence „The three biological replicates of sample were first ground under liquid nitrogen,…“

Line 614: Add a space between the number and the corresponding unit in „30 s“.

Line 632: Add the words „proteins were annotated“ in the sentence „Then proteins were annotated according to the biological process,…“

Line 648: Add a comma preceding the word „respectively“ in the sentence „…known as the „protein network“ and the „chemical universe“, respectively…“

Line 654: Modify the verb „to annotated“ to „to annotate“ in the sentence „First, using KEGG online service tools KAAS …to annotate protein´s KEGG database description.“

Line 670: I do not understand the sentence „Special for prokaryote species (NOT „prokaryon species“), subcellular localization prediction soft CELLO (http://cello.life.nctu.edu.tw/) was used.“ = I think that this sentence should be omitted since no prokaryote species was investigated in the manuscript.

Abbreviations have to be listed alphabetically in Abbreviations list.

Final recommendation: Accept after a minor revision.

Author Response

Mar. 28, 2019

IJMS

We are returning again our revised manuscript (ijms-459555) entitled “Dynamic TMT-based quantitative proteomics analysis of critical initiation process of totipotency during cotton somatic embryogenesis transdifferentiation” that has been revised based on the reviewer's suggestions. We appreciate the time and expertise of the reviewer and the suggestions helped us to further improve our manuscript.

Here are our point by point response to the reviewer's comments:

Reviewer 1

The revised manuscript entitled “Dynamic TMT-based quantitative Proteomics analysis of critical initiation process of totipotency during cotton somatic embryogenesis transdifferentiation” was significantly improved by the authors with respect to the original version.

However, I still have one major comment and several formal comments on the revised manuscript.

1. Major comment: In Table 2 providing basic data on identified proteins revealing differential abundance between PEC vs NEC and GE vs PEC, respectively, proteins revealing differential abundance between GE vs NEC have to be added. I do not understand why proteins revealing differential abundance between GE vs NEC were not included in Table 2. I think that they have to be added there.

RESPONSES: We thank the reviewer’s comment. The data of GE vs NEC Ratio have been added in Table 2. In the original version, in order to focus on comparing the changes of proteins abundance between the neighbor stages, we only showed PEC vs NEC and GE vs PEC. According to the reviewer's suggestion, we supplemented the data of GE vs NEC to make the table more complete.

2. Formal comments:

I still have several formal comments on the revised manuscript.

Results:

Line 312: Correct the typing error in the word“investigated” (not “invesgated”).

Line 317: Modify the term “photosynthesis responsive proteins” (NOT “photosynthesis responded proteins”).

Line 324: Add the verb “are” preceding the verb “involved” in the sentence “…suggested that complex regulatory networks are involved in the cotton SE process…”

Discussion:

Line 506: Correct the typing error in the word “involve” (not “invole”).

Line 523: Correct the verb form “led” (not “leaded”) in the sentence “In the process of carrot SE, the removal of auxin led to the loss of DNA methylation so that the embryo continued to develop…”

Materials and methods,

Line 570: Modify the verb form in the verb “were ground” (not “was grinded”) in the sentence “The three biological replicates of sample were first ground under liquid nitrogen,…”

Line 614: Add a space between the number and the corresponding unit in “30 s“.

Line632: Add the words “proteins were annotated“ in the sentence “Then proteins were annotated according to the biological process,…”

Line 648: Add a comma preceding the word“respectively” in the sentence “…known as the “protein network” and the “chemical universe”, respectively…”

Line 654: Modify the verb “to annotated” to “to annotate” in the sentence “First, using KEGG online service tools KAAS …to annotate protein´s KEGG database description.”

Line 670: I do not understand the sentence “Special for prokaryote species (NOT “prokaryon species”), subcellular localization prediction soft CELLO (http://cello.life.nctu.edu.tw/) was used.” = I think that this sentence should be omitted since no prokaryote species was investigated in the manuscript.

Abbreviations have to be listed alphabetically in Abbreviations list.

RESPONSES: We thank the reviewer for pointing out the formal mistakes there. All the sentences have been revised to be correct accordingly.

Final recommendation: Accept after a minor revision.

We hope you find the revised manuscript satisfactory and thank you for considering our manuscript.

Sincerely,

Fanchang Zeng

Professor

State Key Laboratory of Crop Biology

College of Agronomy, Shandong Agricultural University

Tai’an 271018

Phone: +86-538-8241828

E-mail: fczeng@sdau.edu.cn

Reviewer 2 Report

I appreciate the effort made by the authors in order to improve the manuscript discussion and to address the concerns raised by the review of the first version of this work. After revising the new versions I just have three minor request to authors:

- On figure 9, please change gene for transcript, since even if the dataset is hide, it correspond to a transcriptomic analysis.

- On line 520, include the following text and references (handle reference accordingly to journal guidelines) "...cotton SE transformation as reported in other plants [REF1: https://www.frontiersin.org/articles/10.3389/fpls.2017.00018/full; REF2: https://www.tandfonline.com/doi/pdf/10.1271/bbb.70730; REF3: https://academic.oup.com/treephys/article/26/10/1257/1715621]

- On discussion section, please exchange order of subsection 3.2.4 to 3.2.3 and viceversa. This minor order change will give sense of continuity to the reader and a pulish finish to the manuscript.

Author Response

Mar. 28, 2019

IJMS

We are returning again our revised manuscript (ijms-459555) entitled “Dynamic TMT-based quantitative proteomics analysis of critical initiation process of totipotency during cotton somatic embryogenesis transdifferentiation” that has been revised based on the reviewer's suggestions. We appreciate the time and expertise of the reviewer and the suggestions helped us to further improve our manuscript.

Here are our point by point response to the reviewer's comments:

Reviewer 2

I appreciate the effort made by the authors in order to improve the manuscript discussion and to address the concerns raised by the review of the first version of this work. After revising the new versions I just have three minor request to authors:

1. On figure 9, please change gene for transcript, since even if the dataset is hide, it correspond to a transcriptomic analysis.

RESPONSES: We thank the reviewer and agree to the detailed comments. The term gene” has been changed to “transcript” in the figure 9 as the reviewer suggested.

2. On line 520, include the following text and references (handle reference accordingly to journal guidelines) “...cotton SE transformation as reported in other plants”[REF1: https://www.frontiersin.org/articles/10.3389/fpls.2017.00018/full; REF2: https://www.tandfonline.com/doi/pdf/10.1271/bbb.70730; REF3: https://academic.oup.com/treephys/article/26/10/1257/1715621].

RESPONSES: We thank the reviewer for the valuable comments and suggestions. As the reviewer recommend, we have added the literatures in the corresponding section.

3. On discussion section, please exchange order of subsection 3.2.4 to 3.2.3 and viceversa. This minor order change will give sense of continuity to the reader and a pulish finish to the manuscript.

RESPONSES: We agree with the reviewer for the recommendation. As the reviewer suggestion, we have exchanged order of subsection 3.2.4 to 3.2.3 in the discussion section to make the manuscript more logical with a perfect finish.

We hope you find the revised manuscript satisfactory and thank you for considering our manuscript.

Sincerely,

Fanchang Zeng

Professor

State Key Laboratory of Crop Biology

College of Agronomy, Shandong Agricultural University

Tai’an 271018

Phone: +86-538-8241828

E-mail: fczeng@sdau.edu.cn